# Impact of age and comorbidities on SARS-CoV-2 vaccine-induced T cell immunity

Lisa Loksø Dietz [1,2,17✉], Anna Karina Juhl [1,2,17✉], Ole Schmeltz Søgaard [1,2], Joanne Reekie [3], Henrik Nielsen[4,5], Isik Somuncu Johansen [6,7], Thomas Benfield [8,9], Lothar Wiese[10], Nina Breinholt Stærke [1,2], Tomas Østergaard Jensen [3], Stine Finne Jakobsen [3], Rikke Olesen[2], Kasper Iversen[11], Kamille Fogh[11], Jacob Bodilsen[4,5], Kristine Toft Petersen[4], Lykke Larsen [6,7], Lone Wulff Madsen[6,7], Susan Olaf Lindvig[7], Inge Kristine Holden[7], Dorthe Raben[3], Sidsel Dahl Andersen[1], Astrid Korning Hvidt[1], Signe Rode Andreasen[1], Eva Anna Marianne Baerends[1], Jens Lundgren [3,9,12], Lars Østergaard[1,2], Martin Tolstrup[1,2] & the ENFORCE Study Group*

## Abstract

**Background** Older age and chronic disease are important risk factors for developing severe COVID-19. At population level, vaccine-induced immunity substantially reduces the risk of severe COVID-19 disease and hospitalization. However, the relative impact of humoral and cellular immunity on protection from breakthrough infection and severe disease is not fully understood. **Methods** In a study cohort of 655 primarily older study participants (median of 63 years (IQR: 51–72)), we determined serum levels of Spike IgG antibodies using a Multiantigen Serological Assay and quantified the frequency of SARS-CoV-2 Spike-specific CD4 + and CD8 + T cells using activation induced marker assay. This enabled characterization of suboptimal vaccine-induced cellular immunity. The risk factors of being a cellular hypo responder were assessed using logistic regression. Further follow-up of study participants allowed for an evaluation of the impact of T cell immunity on breakthrough infections. **Results** We show reduced serological immunity and frequency of CD4 + Spike-specific T cells in the oldest age group (≥75 years) and higher Charlson Comorbidity Index (CCI) categories. Male sex, age group ≥75 years, and CCI > 0 is associated with an increased likelihood of being a cellular hypo-responder while vaccine type is a significant risk factor. Assessing breakthrough infections, no protective effect of T cell immunity is identified. **Conclusions** SARS-CoV-2 Spike-specific immune responses in both the cellular and serological compartment of the adaptive immune system increase with each vaccine dose and are progressively lower with older age and higher prevalence of comorbidities. The findings contribute to the understanding of the vaccine response in individuals with increased risk of severe COVID-19 disease and hospitalization.

### Plain language summary

Vaccination has proven very effective in protecting against severe disease and hospitalization of people with COVID-19, the disease caused by SARS-CoV-2. It is still unclear, however, how the different components of the immune system respond to SARS-CoV-2 vaccination and protect from infection and severe disease. Two of the most predominant components of the immune system are specialized proteins and cells. The proteins circulate in the blood and help clear the virus by binding to it, while the cells either kill the virus or help other cells to produce more antibodies. Here, we examined the response of these two components to the SARS-CoV-2 vaccine in 655 Danish citizens. The response of both components was lower in people over 75 years old and with other diseases. These findings help in understanding the immune responses following SARS-CoV-2 vaccination in people at increased risk of severe symptoms of COVID-19.

[1] Department of Infectious Diseases, Aarhus University Hospital, Aarhus, Denmark. [2] Department of Clinical Medicine, Aarhus University, Aarhus, Denmark. [3] Center of Excellence for Health, Immunity and Infections, Rigshospitalet, University of Copenhagen, Copenhagen, Denmark. [4] Department of Infectious Diseases, Aalborg University Hospital, Aalborg, Denmark. [5] Department of Clinical Medicine, Aalborg University, Aalborg, Denmark. [6] Department of Infectious Diseases, Odense University Hospital, Odense, Denmark. [7] Department of Clinical Research, University of Southern Denmark, Odense, Denmark. [8] Department of Infectious Diseases, Copenhagen University Hospital-Amager and Hvidovre, Hvidovre, Denmark. [9] Department of Clinical Medicine, University of Copenhagen, Copenhagen, Denmark. [10] Department of Medicine, Zealand University Hospital, Roskilde, Denmark. [11] Department of Cardiology and Department of Emergency Medicine, Herlev Hospital, Herlev, Denmark. [12] Dept of Infectious Diseases, Copenhagen University Hospital-Rigshospitalet, Copenhagen, Denmark. [17] These authors contributed equally: Lisa Loksø Dietz, Anna Karina Juhl. *A list of authors and their affiliations appears at the end of the paper. ✉email: lisdie@rm.dk; anajuh@rm.dk

In December 2019, the outbreak of severe acute respiratory syndrome coronavirus 2 (SARS-CoV-2) started the COVID-19 pandemic[1]. COVID-19 disease severity varies from asymptomatic or mild symptoms to severe acute respiratory disease which can be fatal[2]. In particular, the older population and people with chronic diseases are at a higher risk of developing severe COVID-19[3,4]. However, the introduction of vaccination programs against SARS-CoV-2 have proven an effective health measure against severe disease and mortality[5–7].

Findings in animal studies have shown contributions from both the humoral and cellular immune response on protection from SARS-CoV-2 infection. It is therefore important to understand both compartments following vaccination against SARS-CoV-2. B cells and antibodies are essential components of the immunological memory against respiratory infections. Hence, most SARS-CoV-2 mRNA vaccine studies have focused on characterizing the post-immunization humoral response[8–12]. Previous studies found a correlation between antibody levels and protection against COVID-19 caused by the original Wuhan-B strain as well as Alpha and Delta variants[13–15]. Additionally, observational studies have shown that SARS-CoV-2 mRNA vaccines induce cellular immunity evidenced by SARS-CoV-2 Spike-specific CD4 + and CD8 + T cells that persist for at least 6 months post-immunization, with the magnitude of CD4 + cells exceeding that of CD8 + T cells[16–18]. In convalescent individuals, a potent SARS-CoV-2 Spike-specific CD4 + and CD8 + T cell response has also been reported. Such responses have been associated with decreased disease severity[19]. Older age is known to be associated with diminished vaccine efficiency[20], but COVID-19 vaccine development and phase III efficacy has only been assessed in a small proportion of elderly individuals[10,16,17,21,22]. Hence, further knowledge is needed on this subject.

In the present study, we profile serological and cellular immunity following SARS-CoV-2 vaccination in a cohort of 655 participants with a high representation of elderly individuals with a pronounced burden of comorbidities. We investigate the association between breakthrough infections and T cell immunity. Finally, sex, age, CCI, and vaccine type are evaluated as predictors of SARS-CoV-2 vaccine response.

## Methods

The National Cohort Study of Effectiveness and Safety of SARS-CoV-2 vaccines (ENFORCE) is a Danish open-label Phase IV study, which is non-randomized with parallel groups. The study enrolled Danish citizens prior to vaccination against COVID-19 (clinicaltrials.gov, identifier: NCT04760132). An article on serological immune response and durability on the entire ENFORCE cohort has been published[4]. The present study was a predefined ENFORCE T cell immunity substudy, which was part of the master protocol of the ENFORCE study. The current study reports on interim results for work packages 3 and 4, which has been approved by the ENFORCE steering committee. The primary objective of the substudy is determination of cellular immunity following COVID-19 vaccination among a subset of ENFORCE participants.

**Study population and data collection**. Danish citizens from all five Danish regions, aged 18 years or older, and scheduled to receive a COVID-19 vaccine were included. The ENFORCE T cell immunity substudy aimed to enroll 10% of the total ENFORCE participants.

Inclusion criteria were (1) Signed informed consent, (2) 18 years of age or above eligible to receive a SARS-CoV-2 vaccine, and (3) Willingness to comply with trial protocol (including follow-up visits and biological samples). Exclusion criteria were (1) Individuals under the age of 18, (2) Individuals for which the vaccines are contraindicated, and (3) Previous SARS-CoV-2 vaccination.

In the present study, participants with a SARS-CoV-2 infection prior to baseline, defined as positive SARS-CoV-2 PCR test or Spike Ig positive at baseline visit (WANTAI assay), were excluded from the analysis.

Information on age, sex, medical history, vaccination dates, and vaccine type (BTN162b2, mRNA-1273 and ChAdOx1) was obtained from the Danish National Patient Registry (DNPR) and the Danish Vaccination Registry.

Data from all SARS-CoV-2 PCR tests were acquired from the national Key Infectious Diseases System (KIDS) database, and specific variant information was available from the Danish Microbiology Database (MiBa).

The study protocol was approved by the Danish Medicines Agency (#2020-006003-42), and the National Committee on Health Research Ethics (#1-10-72-337-20). All participants provided informed written consent.

**Data on comorbidity**. Information on comorbidity was based on each individual medical history within 5 years prior to study entry date as described in Søgaard et al.[4], this data was obtained from the DNPR. The CCI is a validated measure of comorbidity[23]. The CCI score is calculated by assigning weights to each of 17 major disease categories[24]. Three categories of comorbidity were defined based on the CCI score; low (CCI = 0), medium (CCI = 1–2), and high (CCI > 2).

**Follow-up**. The second study visit (study visit day 21) occurred 0–7 days prior to the second vaccine dose (median of 21, 34, and 84 days after the first vaccine dose for BTN162b2, mRNA-1273, and ChAdOx1, respectively). The third study visit (study visit day 90) occurred 90 days (+/−14 days) after the first vaccine dose (a median of 91 days for BTN162b2 and mRNA-1273, and 100 days for ChAdOx1). Blood samples for measuring SARS-CoV-2 Spike-specific T cells and IgG levels were obtained at each study visit (Supplementary Fig. 1).

**Sample collection**. Of the 655 participants included in the study, blood samples for analysis of SARS-CoV-2 Spike antibody profiling were collected for 560 (99.2%), 598 (91.3%), and 523 (79.8%) participants at baseline (day 0), day 21, and day 90, respectively. Blood samples for analysis of SARS-CoV-2 Spike-specific T cells with the activation induced markers (AIM) assay were collected for 638 (97.4%), 573 (87.5%), and 502 (76.6%) participants at baseline (day 0), day 21, and day 90, respectively. Logistic challenges in the beginning of study enrollment, led to poor sample quality for a number of baseline samples for analysis with the AIM assay. As most participants enrolled early in the study received the ChAdOx1 vaccine, all baseline samples in this vaccine group were lost. Thus, 286 and 272 of the 638 baseline samples met the quality criteria set for the AIM assay for CD4 + or CD8 + Spike-specific T cells, respectively. Of the 573 samples analysed at day 21, 460 (CD4 + T cells) and 444 (CD8 + T cells) met the quality criteria. Lastly, of the 502 samples at the day 90 visit, 462 (CD4 + T cells) and 449 (CD8 + T cells) met quality criteria. Samples that did not meet quality criteria were excluded from all data analysis.

**Peripheral blood mononuclear cell (PBMC) isolation**. Whole blood was collected in sodium citrate/Ficoll blood collection tubes from BD Vacutainer (BD CPT, Cat. No.: BDAM362782). PBMCs were isolated from three CPT tubes per participant. CPT

tubes were centrifuged at 1500xg for 20 mins within 2 h of blood collection. Centrifuged CPT tubes were reverted slowly, the supernatant of all three CPT tubes were pooled and centrifuged at 400xg for 10 mins. The supernatant was then discarded and PBMCs were resuspended and washed in PBS containing 2% FBS. PBMCs were pelleted by centrifugation at 400xg for 10 mins and resuspended in media for cryopreservation (FBS with 10% DMSO). Immediately following resuspension in media with DMSO, cells were placed in freezing containers and cryopreserved at −80 °C for at least 24 h, subsequently the cells were moved to −150 °C for long-term storage.

**SARS-CoV-2 Spike-specific T cells**. The percentage of SARS-CoV-2 Spike-specific T cells was measured at all three study visits (day 0, day 21, and day 90) using the AIM assay. The assay is based on detection of activation induced markers as a measure of antigen specific cells[25,26]. Antigen specific T cells were defined as cells that express 2 or 3 activation induced markers. In the present study, the AIMs were CD69, OX40 (CD134), and 41BB (CD137).

Purified PBMCs were stimulated with PepMix™ SARS-CoV-2 (JPT peptides product code PM-WCPV-S-2) at 2 μg/ml or negative control (Dimethyl sulfoxide) for 20 h. The PepMixTM contains a pool of 315 (158 + 157) peptides derived from a peptide scan (15mers with 11 aa overlap) through the Spike glycoprotein (Swiss-Prot ID: P0DTC2) of SARS-CoV-2 (Wuhan-Hu-1 lineage). Following stimulation, cells were washed, stained and run on MACSQuant16 (Miltenyi Biotec). Data was analyzed using FlowJo™ v10.8.1 Software (BD Biosciences).

For viability staining 0.1 μl LIVE/DEAD-APC-H7 (cat# L34976) was diluted in 99.9 μl PBS. The staining master mix contained CD3-PerCP-Cy5.5 (1 μl, cat# BL344808), CD4-BV650 (2 μl, cat# BL300536), CD8-BV605 (1 μl, cat# BL301040), CD69-APC (1 μl, cat# BL310910), OX40-BV421 (5 μl, cat# BL 350014), and 41BB-PE (2.5 μl, cat# BL309804) in 52.5 μl Brilliant Stain Buffer (cat# 566349).

Live cells were gated by the dead cell stain as the negative population. Single cells were gated in a FSC-A/FSC-H plot. Lymphocytes were gated in a FSC-A/SSC-A plot. CD3 + cells (T lymphocytes) were gated by CD3 positivity. CD4 + and CD8 + T cells were gated as single positive for either CD4 or CD8, respectively. Lastly both CD4 + and CD8 + cells were gated for the three AIMs (Supplementary Fig. 2). Boolean gating for the three AIMs was performed on both CD4 + and CD8 + T cells to identify double- and triple positive cells. Lastly, background was subtracted from Spike stimulated cells to get the final percentage of SARS-CoV-2 Spike-specific T cells. Samples with negative values (i.e., where background signal was higher than signal in Spike stimulated cells) were turned to zero.

Data was excluded if either the viability of the sample was below 70% at flow data acquisition or if the CD4 + and/or CD8 + T cell count was below 10,000.

**SARS-CoV-2 antibody profiling**. Serum levels of SARS-CoV-2 Spike IgG antibodies were measured at all study visits using the MesoScale Diagnostic Multiantigen Serology Assay according to manufacturer's instructions. Samples were diluted 1:5000 and the plates were read on a MESO SECTOR S600 Reader. Data analysis was performed utilizing the MSD Discovery Workbench Software (Version 4.0). Total IgG concentrations were calculated by fitting the electro chemiluminescence signals to the calibration curves.

**Serological and cellular vaccine responder group**. The serological vaccine responder group was defined as in Søgaard et al.[4] based on the participants change in Spike IgG at day 90 relative to their pre-vaccine (baseline) levels: vaccine hypo-responders were individuals who had <2 $\log_{10}$ fold change in Spike IgG, moderate

responders were individuals with a 2–3 $\log_{10}$ fold change, and high responders with a > 3 $\log_{10}$ fold change in Spike IgG.

The study participants were stratified into cellular vaccine responder groups (hypo-responders and responders) to assess the cellular responsiveness with the AIM assay, and define characteristics of participants with a low- or absent AIM response (hypo-responders) vs. a higher AIM response (responders). Since most baseline values for cellular immunity were zero, calculation of a fold change from baseline to day 90 was not feasible. Thus, the cellular vaccine responder group was instead defined by a threshold value for the percentage of SARS-CoV-2 Spike-specific T cells at peak cellular immunity (day 90). This threshold value was set at the median value at baseline plus one standard deviation (SD) resulting in a value of 0.107% and 0.078% for SARS-CoV-2 Spike-specific CD4 + T cells and CD8 + T cells, respectively.

**Data on breakthrough infections and SARS-CoV-2 variants**. A breakthrough infection was defined as a positive SARS-CoV-2 PCR test result occurring after day 90 with follow-up censored at the date of last database update (March 7th 2022). Virus variant information was available from either variant PCR results or whole genome sequencing. When no viral subtyping was available, the most likely variant was defined by sample date based on when specific variants were most prevalent in Denmark. Breakthrough infections with missing subtyping information between July 1st 2021 and December 1st 2021 were considered SARS-CoV-2 lineage B.1.617.2 (Delta variant), samples collected after the 21st of December 2021 were SARS-CoV-2 lineage B.1.1.529 (Omicron variant) and samples collected between the 1st and 21st of December 2021 were unknown variant.

**Statistics and reproducibility**. Baseline demographics and clinical characteristics of participants were tabulated showing medians with upper- and lower quartiles [Q1, Q3] or $n$ and percentages. $P$-values in all tables were calculated by unpaired, non-parametric Kruskal–Wallis test (two-tailed) for continuous variables and Chi-squared test or Fishers exact test (sample size < 5) test for categorical variables.

Boxplots were used to present the T cell- and serology data showing the quartiles of the dataset. Whiskers extended to show the rest of the distribution within 1.5 times the interquartile range (IQR). Outliers beyond 1.5 times the IQR were not shown. Data was compared using unpaired, non-parametric Mann–Whitney $U$-test (two-tailed) with Bonferroni correction. $P$-values ≤ 0.05 are considered significant; when nothing is shown, results are non-significant ($p$-value > 0.05).

Correlations between CD4 + and CD8 + T cell SARS-CoV-2 Spike-specificity was determined using Spearman's correlation coefficient ($\rho$). Correlation strength was interpreted using Mukaka, M. M. (2012)[27]. Data was plotted with a linear regression model fit where translucent bands showed the 99% confidence interval estimated using a bootstrap.

Risk factors for cellular vaccine hypo-responsiveness at day 90 were investigated by multivariable logistic regression including sex, age group, CCI score, and vaccine type. All predictor variables were selected as described in Søgaard et al.[4].

All data analysis and visualization was done using Python version 3.9 using matplotlib and seaborn[28,29], except for logistic regression which was done using the generalized linear model from R stats version 3.6.2[30]. Tables were created with tableone version 0.7.10 in Python[31].

**Reporting summary**. Further information on research design is available in the Nature Portfolio Reporting Summary linked to this article.

## Results

A total of 699 participants were enrolled in the ENFORCE T cell immunity substudy (see Methods). Due to prior SARS-CoV-2 infection, 44 were excluded. Thus, the study cohort consisted of 655 participants (56.3% females) with a median age of 63 years (IQR: 51–72). The majority received two doses of BTN162b2 (46.8%, $n = 314$), or two doses of mRNA-1273 (37.4%, $n = 251$), while 15.8% ($n = 106$) received one dose of ChAdOx1 followed by a second dose of either of the two mRNA vaccines. Individuals who received BTN162b2 had a higher prevalence of comorbidities and a higher median age (71 years) than both mRNA-1273 and ChAdOx1 recipients (median ages of 62 and 50 years, respectively). Moreover, participants receiving ChAdOx1 were predominantly young female healthcare workers with very few comorbidities (Table 1).

**Dynamics of SARS-CoV-2 Spike-specific CD4 + and CD8 + T cells following vaccination.** SARS-CoV-2 Spike-specific T cell immunity was assessed for both CD4 + and CD8 + T cells using the Activation Induced Marker (AIM) assay. The frequency of CD4 + and CD8 + Spike-specific T cells increased significantly after the first and second vaccination. For Spike-specific CD4 + T cells, the median percentage at baseline (day 0) increased significantly from 0% (IQR: 0.00–0.08, $n = 286$) to 0.26% (IQR: 0.07–0.52, $n = 460$) after the first vaccine dose (day 21) and further to 0.43% (IQR: 0.23–0.74, $n = 462$) following the second vaccine dose (day 90) (Fig. 1a). Albeit at a lower magnitude, significant changes were also observed for CD8 + Spike-specific T cells throughout the study visits increasing from 0.02% (IQR: 0.00–0.04, $n = 272$) at baseline to 0.12% (IQR: 0.05–0.23, $n = 444$) and 0.17% (IQR: 0.07–0.34, $n = 449$) pre- and post-second vaccine dose, respectively (Fig. 1b).

Stratifying individuals by vaccine type revealed a similar pattern of CD4 + and CD8 + Spike-specific T cell dynamics (Supplementary Fig. 3), with significant increases for both mRNA vaccine groups from baseline to day 21, and from day 21 to day 90. No baseline data could be obtained from the ChAdOx1 vaccine group (see Methods), however, an increase in Spike CD4 + and CD8 + T cells was observed from day 21 to 90, though this increase was not statistically significant (Supplementary Fig. 3).

Further, the Spike-specific responses in the two T cell lineages correlated positively, $rho = 0.54$ at day 21 and $rho = 0.64$ at day 90 (Supplementary Fig. 4).

**Table 1 Participant demographics at study enrollment by vaccine type.**

|  | Vaccine type | | | | |
|---|---|---|---|---|---|
|  | Total ($n = 655$) | BNT162b2 ($n = 306$) | mRNA-1273 ($n = 247$) | ChAdOx1+mRNA ($n = 102$) | p-value |
| Age at enrollment, median [Q1,Q3] | 63 [51,72] | 71 [55,77] | 62 [51,68] | 50 [33,58] | <0.001 |
| Age group, n (%) |  |  |  |  |  |
| <65 | 353 (53.9) | 104 (34.0) | 147 (59.5) | 102 (100.0) | <0.001 |
| 65–74 | 157 (24.0) | 75 (24.5) | 82 (33.2) |  |  |
| ≥75 | 145 (22.1) | 127 (41.5) | 18 (7.3) |  |  |
| Sex, n (%) |  |  |  |  |  |
| Male | 286 (43.7) | 157 (51.3) | 117 (47.4) | 12 (11.8) | <0.001 |
| Female | 369 (56.3) | 149 (48.7) | 130 (52.6) | 90 (88.2) |  |
| Vaccine-priority group, n (%) |  |  |  |  |  |
| Individuals at increased risk | 194 (29.9) | 165 (54.3) | 29 (11.9) |  | <0.001 |
| General population | 329 (50.7) | 118 (38.8) | 211 (86.8) |  |  |
| Health-care workers | 126 (19.4) | 21 (6.9) | 3 (1.2) | 102 (100.0) |  |
| CCI, n (%) |  |  |  |  |  |
| 0 | 497 (75.9) | 191 (62.4) | 207 (83.8) | 99 (97.1) | <0.001 |
| 1–2 | 123 (18.8) | 85 (27.8) | 35 (14.2) | 3 (2.9) |  |
| >2 | 35 (5.3) | 30 (9.8) | 5 (2.0) |  |  |
| Comorbidities in the previous 5 years, n |  |  |  |  |  |
| Myocardial infarction | 8 | <5* | 5 |  | 0.346 |
| Congestive heart failure | 18 | 13 | 5 |  | 0.049 |
| Peripheral vascular disease | 6 | 5 | <5* |  | 0.268 |
| Cerebrovascular disease | 20 | 13 | 7 |  | 0.080 |
| Dementia |  |  |  |  |  |
| Chronic pulmonary disease | 30 | 19 | 9 | <5* | 0.167 |
| Rheumatological disease | 13 | 9 | <5* | <5* | 0.377 |
| Peptic ulcer disease | <5* | <5* | <5* |  | 1.000 |
| Mild liver disease | 14 | 13 | <5* |  | 0.002 |
| Diabetes without chronic complications | 20 | 13 | 7 |  | 0.080 |
| Diabetes with chronic complications | <5* | <5* | <5* |  | 1.000 |
| Hemiplegia or paraplegia |  |  |  |  |  |
| Any malignancy, including leukemia and lymphoma | 82 | 62 | 20 |  | <0.001 |
| Moderate or severe liver disease | 5 | <5* | <5* |  | 0.471 |
| Metastatic solid tumor | <5* | <5* |  |  | 0.166 |
| AIDS/HIV | 11 | 8 | <5* |  | 0.196 |
| Renal disease | 10 | 10 |  |  | 0.003 |
| Organ transplantation | 38 | 35 | <5* |  | <0.001 |

*Groups with small numbers (<5 participants per cell) where there is the potential that individual participants could be identified or be able to identify themselves have been edited to maintain participant confidentiality.

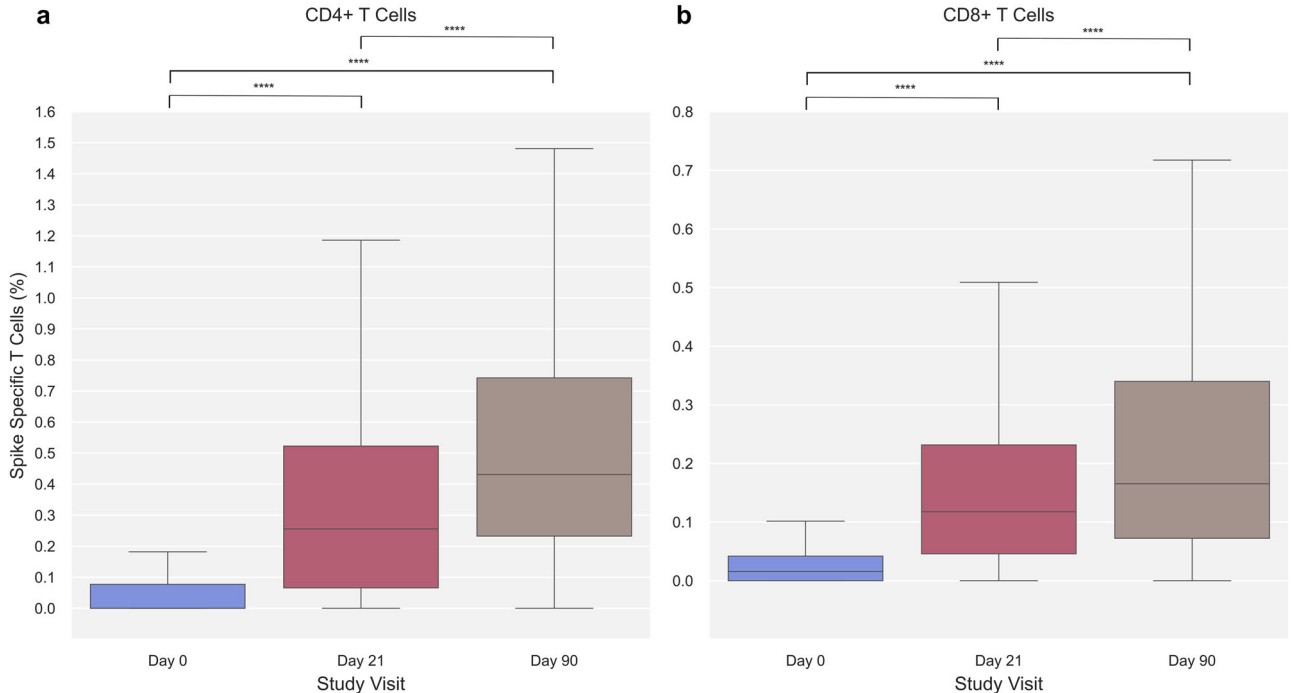

**Fig. 1 SARS-CoV-2 Spike-specific T cells at each study visit. a** SARS-CoV-2 Spike-specific CD4 + T cells at day 0 (blue), 21 (red), and 90 (brown) (n = 286, 460, and 462, respectively). **b** SARS-CoV-2 Spike-specific CD8 + T cells at day 0 (blue), 21 (red), and 90 (brown) (n = 272, 444, and 449, respectively). Data was compared using unpaired, non-parametric Mann–Whitney U-test. Error bars show the distribution within 1.5 times IQR. p-value annotation legend: *$1.00 \cdot 10^{-02} < p \leq 5.00 \cdot 10^{-02}$, **$1.00 \cdot 10^{-03} < p \leq 1.00 \cdot 10^{-02}$, ***$1.00 \cdot 10^{-04} < p \leq 1.00 \cdot 10^{-03}$, ****$p \leq 1.00 \cdot 10^{-04}$.

**Cellular and serological vaccine-induced immunity with increasing age and comorbidity burden.** As increasing age and certain chronic diseases are risk factors for severe COVID-19 disease, the impact of age and CCI on cellular and serological vaccine-induced immunity was assessed.

Stratifying participants according to age (<65, 65–74, ≥75 years) revealed a lower frequency of Spike-specific CD4 + T cells post-vaccination in older age groups. After the first vaccination (day 21), the percentage of Spike-specific CD4 + T cells was 0.33% (IQR: 0.11–0.60, n = 234) in the youngest age group (<65 years) compared to 0.26% (IQR: 0.07–0.52, n = 123) and 0.12% (IQR: 0.00–0.35, n = 103) in age groups 65–74 and ≥75 years, respectively. After the second vaccination (day 90) the corresponding values were 0.51% (IQR: 0.29–0.75, n = 214), 0.45% (IQR: 0.25–0.77, n = 134) and 0.29% (IQR: 0.10–0.62, n = 114) (Fig. 2a).

Of note, the median percentage of Spike-specific CD4 + T cells of the oldest age group at the post-second vaccine (day 90) time point (0.29%, IQR: 0.10-0.62) was equivalent to the median value of the youngest age group (0.33%, IQR: 0.11–0.60) at the pre-second vaccine (day 21) time-point. A similar pattern was seen for CD8 + T cells, though of less magnitude and not significant (Fig. 2b).

Further, the percentage of day 90 CD4 + Spike-specific T cells was lower in participants with higher CCI (Fig. 2c, d and Supplementary Fig. 5). This was most pronounced in participants aged <65 and 65–74, compared to ≥75 years.

Serum levels of SARS-CoV-2 Spike IgG antibodies were also assessed in participants stratified by age and CCI score. Generally, similar patterns were observed for the levels of Spike IgG compared to Spike-specific T cells; the level of Spike IgG increased after both the first and second vaccine dose across all age groups. Additionally, stratifying participants by age revealed a reduced level of Spike IgG in older age groups following both first and second vaccine dose (Fig. 3a). Furthermore, when stratifying

participants according to CCI score, the day 90 Spike IgG levels were progressively lower in participants with higher CCI (Fig. 3b).

**Relation between serological and cellular vaccine-induced immunity.** To determine the connection between cellular and serological vaccine-induced immunity, differences in cellular immunity between three serological vaccine responder groups were assessed.

A serological vaccine responder group could be assigned to 498 participants with data on SARS-CoV-2 Spike IgG both at baseline and day 90. Of the 498 participants, 44 (8.8%) were hypo-, 114 (22.9%) moderate-, and 340 (68.3%) high-responders. The median $\log_{10}$ fold change in SARS-CoV-2 Spike IgG was 1.34 (hypo-responders), 2.77 (moderate-responders), and 3.6 (high-responders).

A significantly increased level of both CD4 + and CD8 + Spike-specific T cells at day 90 was observed among serological high responders (0.52% and 0.19%, respectively) compared to moderate- (0.27% and 0.15%, respectively) and hypo-responders (0.30% and 0.06%, respectively). No significant difference in cellular immune response was observed between participants classified in the hypo- and moderate responder groups (Fig. 4).

Additionally, tertile stratification of day 90 CD4 + and CD8 + T cell response revealed a significant increase in the amount of SARS-CoV-2 Spike IgG antibodies at day 90 from the lowest to the middle and to the highest tertile ($1.15 \cdot 10^5$ AU/mL, $3.03 \cdot 10^5$ AU/mL, and $4.04 \cdot 10^5$ AU/mL, respectively, for CD4 + and $1.30 \cdot 10^5$ AU/mL, $3.16 \cdot 10^5$ AU/mL, and $3.95 \cdot 10^5$ AU/mL, respectively, for CD8 + ) (Supplementary Table 1).

**Impact of vaccine-induced cellular immunity on breakthrough infections.** As reduced levels of vaccine-induced immunity may increase the risk of breakthrough infections, the study sought to

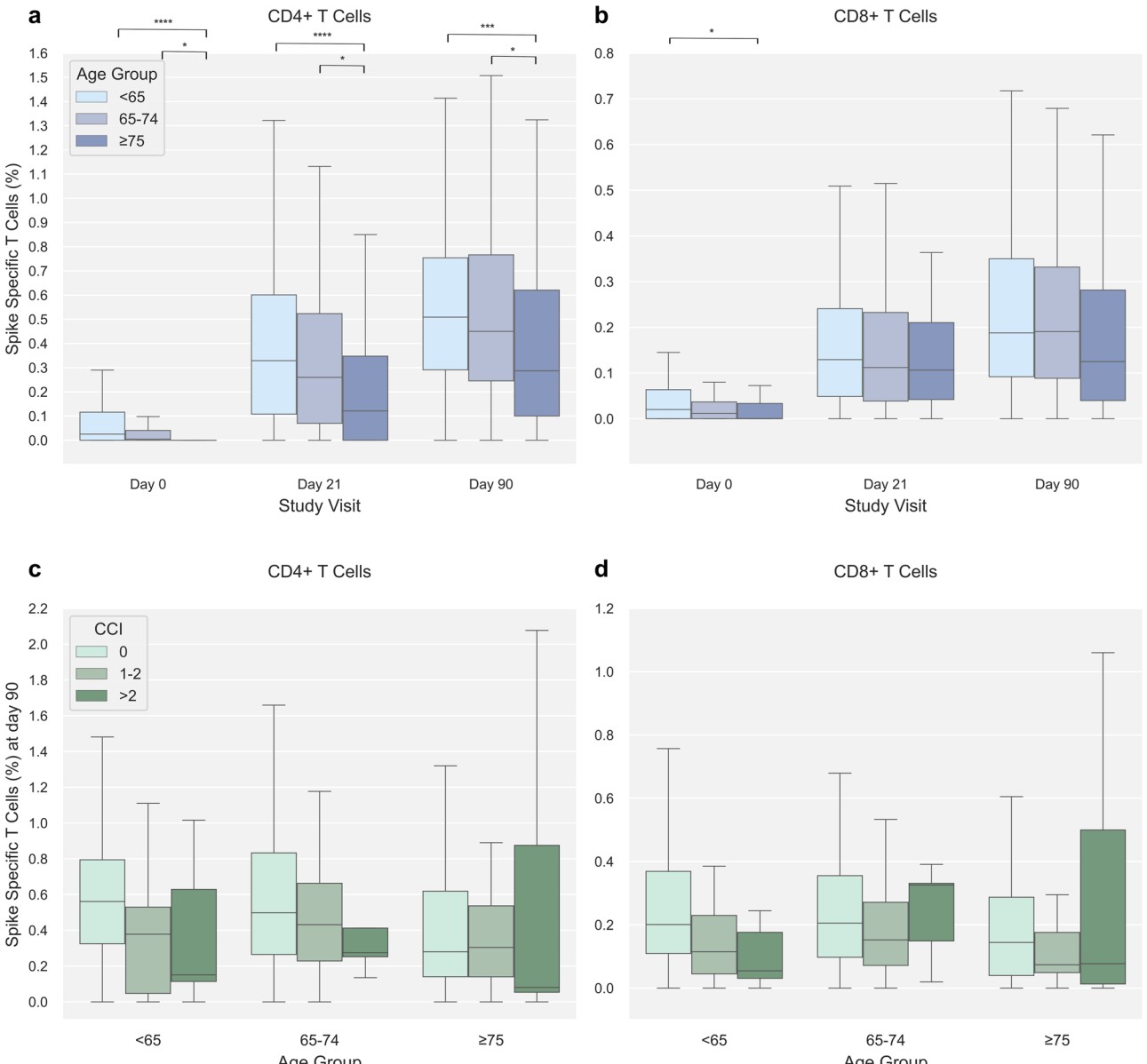

**Fig. 2 SARS-CoV-2 Spike-specific T cells with age and Charlson Comorbidity Index (CCI). a** SARS-CoV-2 Spike-specific CD4 + T cells at 0, 21, and 90 stratified by age group (<65 years [light blue], 65–74 years [blue], ≥75 years [dark blue]). Group sizes are; at day 0 n = 142 (<65 years), n = 54 (65–74 years), n = 90 (≥75 years); at day 21 n = 234 (<65 years), n = 123 (65–74 years), n = 103 (≥75 years); at day 90 n = 214 (<65 years), n = 134 (65–74 years), n = 114 (≥75 years). **b** SARS-CoV-2 Spike-specific CD8 + T cells at 0, 21, and 90 stratified by age group (<65 years [light blue], 65–74 years [blue], ≥75 years [dark blue]). Group sizes are; at day 0 n = 139 (<65 years), n = 54 (65–74 years), n = 79 (≥75 years); at day 21 n = 233 (<65 years), n = 118 (65–74 years), n = 93 (≥75 years); at day 90 n = 213 (<65 years), n = 130 (65–74 years), n = 106 (≥75 years). **c** SARS-CoV-2 Spike-specific CD4 + T cells at day 90 in the three age groups stratified by Charlson Comorbidity Index (CCI) (CCI = 0 [light green], CCI = 1–2 [green], CCI > 2 [dark green]). Group sizes are; in age group <65 years n = 175 (CCI = 0), n = 30 (CCI = 1–2), n = 9 (CCI > 2); age group 65–74 years n = 88 (CCI = 0), n = 37 (CCI = 1–2), n = 5 (CCI > 2); age group ≥75 years n = 79 (CCI = 0), n = 21 (CCI = 1–2), n = 6 (CCI > 2). **d** SARS-CoV-2 Spike-specific CD8 + T cells at day 90 in the three age groups stratified by CCI (CCI = 0 [light green], CCI = 1–2 [green], CCI > 2 [dark green]). Group sizes are; in age group <65 years n = 175 (CCI = 0), n = 29 (CCI = 1–2), n = 9 (CCI > 2); age group 65–74 years n = 88 (CCI = 0), n = 37 (CCI = 1–2), n = 5 (CCI > 2); age group ≥75 years n = 79 (CCI = 0), n = 21 (CCI = 1–2), n = 6 (CCI > 2). Data was compared using unpaired, non-parametric Mann–Whitney U-test. Error bars show the distribution within 1.5 times IQR. p-value annotation legend: *1.00·10$^{-02}$ < p ≤ 5.00·10$^{-02}$, **1.00·10$^{-03}$ < p ≤ 1.00·10$^{-02}$, ***1.00·10$^{-04}$ < p ≤ 1.00·10$^{-03}$, ****p ≤ 1.00·10$^{-04}$.

examine whether day 90 vaccine-induced immunity could affect occurrence of documented breakthrough infections during the observation period ending March 7th 2022.

Following their day 90 study visit participants were followed-up for a median of 238 days. A documented breakthrough infection was observed in 136 (29.4%) and 134 (29.8%) of participants with available Spike-specific T cell data for CD4 + (n = 462) and CD8 + (n = 449) T cells, respectively, at day 90. None of the breakthrough infection cases led to severe disease or hospitalization. Stratifying participants by breakthrough infection status, showed no significant difference in cellular response at day 90 for either CD4 + or CD8 + T cells (Supplementary Fig. 6a, c). Further, tertile stratification of day 90 CD4 + and CD8 + T cell response revealed no significant difference in the distribution of breakthrough infections across the three tertiles for neither CD4 + nor CD8 + T cells

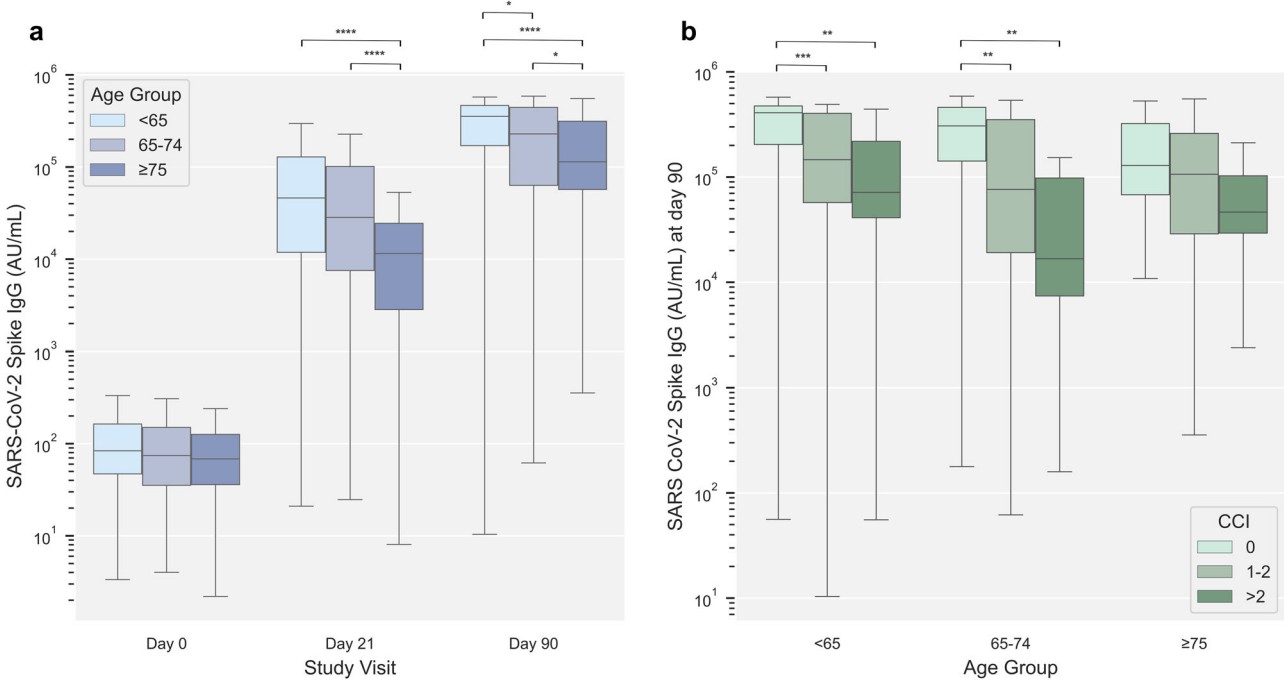

**Fig. 3 SARS-CoV-2 Spike IgG by age and Charlson Comorbidities Index (CCI). a** SARS-CoV-2 Spike IgG arbitrary units (AU)/ml at day 0, 21, and 90 stratified by age group (<65 years [light blue], 65–74 years [blue], ≥75 years [dark blue]). Group sizes are; at day 0 $n = 352$ (<65 years), $n = 154$ (65–74 years), $n = 144$ (≥75 years); at day 21 $n = 304$ (<65 years), $n = 152$ (65–74 years), $n = 142$ (≥75 years); at day 90 $n = 238$ (<65 years), $n = 146$ (65–74 years), $n = 139$ (≥75 years). **b** SARS-CoV-2 Spike IgG at day 90 in the three age groups stratified by Charlson Comorbidity Index (CCI) (CCI = 0 [light green], CCI = 1–2 [green], CCI > 2 [dark green]). Group sizes are; in age group <65 years $n = 195$ (CCI = 0), $n = 33$ (CCI = 1–2), $n = 10$ (CCI > 2); age group 65–74 years $n = 92$ (CCI = 0), $n = 46$ (CCI = 1–2), $n = 8$ (CCI > 2); age group ≥ 75 $n = 97$ (CCI = 0), $n = 32$ (CCI = 1–2), $n = 10$ (CCI > 2). Data was compared using unpaired, non-parametric Mann–Whitney U-test. Error bars show the distribution within 1.5 times IQR. p-value annotation legend: *$1.00 \cdot 10^{-02} < p \leq 5.00 \cdot 10^{-02}$, **$1.00 \cdot 10^{-03} < p \leq 1.00 \cdot 10^{-02}$, ***$1.00 \cdot 10^{-04} < p \leq 1.00 \cdot 10^{-03}$, ****$p \leq 1.00 \cdot 10^{-04}$.

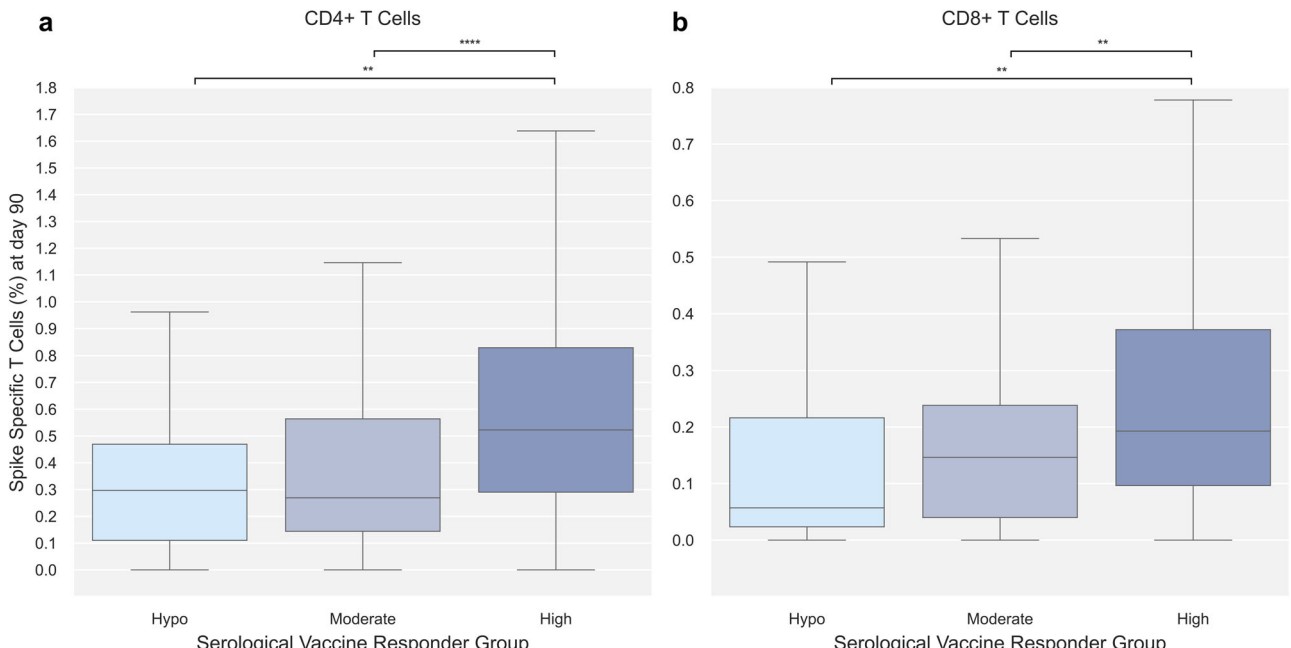

**Fig. 4 SARS-CoV-2 Spike-specific T cells at day 90 by serological vaccine responder group.** Day 90 SARS-CoV-2 Spike-specific T cells stratified by serological vaccine responder group (Hypo [light blue], Moderate [blue], High [dark blue]). Serological vaccine responder group was defined based on the change in Spike IgG at day 90 relative to their pre-vaccine (baseline) levels: vaccine hypo-, moderate- and high responders were individuals with a $\log_{10}$ fold change of <2, 2-3, or >3 in Spike IgG, respectively. **a** SARS-CoV-2 Spike-specific CD4 + T cells; $n = 46$ (Hypo), $n = 105$ (Moderate), $n = 318$ (High). **b** SARS-CoV-2 Spike-specific CD8 + T cells; $n = 36$ (Hypo), $n = 101$ (Moderate), $n = 309$ (High). Data was compared using unpaired, non-parametric Mann–Whitney U-test. Error bars show the distribution within 1.5 times IQR. p-value annotation legend: *$1.00 \cdot 10^{-02} < p \leq 5.00 \cdot 10^{-02}$, **: $1.00 \cdot 10^{-03} < p \leq 1.00 \cdot 10^{-02}$, ***$1.00 \cdot 10^{-04} < p \leq 1.00 \cdot 10^{-03}$, ****$p \leq 1.00 \cdot 10^{-04}$.

**Table 2 Vaccine responder group for CD4 + and CD8 + T cells at day 90.**

| | Day 90 CD4 + response group | | | Day 90 CD8 + response group | | |
|---|---|---|---|---|---|---|
| | **Responder** | **Hypo-responder** | **p-value** | **Responder** | **Hypo-responder** | **p-value** |
| Number of participants, n (%) | 399 (86.4) | 63 (13.6) | | 330 (73.5) | 119 (26.5) | |
| **Baseline characteristics** | | | | | | |
| Age at enrollment, median [Q1,Q3] | 66 [52,71] | 72 [58,78] | 0.002 | 64 [52,70] | 68 [54,76] | 0.015 |
| Age group, n (%) | | | | | | |
| <65 | 192 (89.7) | 22 (10.3) | <0.001 | 166 (77.9) | 47 (22.1) | 0.001 |
| 65–74 | 123 (91.8) | 11 (8.2) | | 101 (77.7) | 29 (22.3) | |
| ≥75 | 84 (73.7) | 30 (26.3) | | 63 (59.4) | 43 (40.6) | |
| Sex, n (%) | | | | | | |
| Male | 186 (83.0) | 38 (17.0) | 0.059 | 154 (71.6) | 61 (28.4) | 0.451 |
| Female | 213 (89.5) | 25 (10.5) | | 176 (75.2) | 58 (24.8) | |
| Vaccine type, n (%) | | | | | | |
| BNT162b2 | 184 (76.7) | 56 (23.3) | <0.001 | 130 (57.0) | 98 (43.0) | <0.001 |
| mRNA-1273 | 208 (97.7) | 5 (2.3) | | 194 (91.5) | 18 (8.5) | |
| ChAdOx1+mRNA | 7 (77.8) | 2 (22.2) | | 6 (66.7) | 3 (33.3) | |
| Days between first and second dose, median [Q1,Q3] | 33 [22,35] | 23 [21,27] | <0.001 | 35 [22,35] | 23 [21,28] | <0.001 |
| Days from first vaccine to third study visit, median [Q1,Q3] | 91 [88,95] | 91 [89,92] | 0.922 | 91 [88,95] | 91 [88,95] | 0.348 |
| CCI, n (%) | | | | | | |
| 0 | 313 (89.2) | 38 (10.8) | 0.004 | 264 (77.2) | 78 (22.8) | 0.005 |
| 1–2 | 72 (79.1) | 19 (20.9) | | 55 (63.2) | 32 (36.8) | |
| >2 | 14 (70.0) | 6 (30.0) | | 11 (55.0) | 9 (45.0) | |
| Comorbidities in the previous 5 years, n | | | | | | |
| Myocardial infarction | 5 | <5* | 0.587 | <5* | <5* | 0.193 |
| Congestive heart failure | 11 | <5* | 0.694 | 8 | 5 | 0.343 |
| Peripheral vascular disease | <5* | <5* | 0.521 | <5* | <5* | 1.000 |
| Cerebrovascular disease | 16 | <5* | 0.733 | 15 | <5* | 0.423 |
| Dementia | | | | | | |
| Chronic pulmonary disease | 11 | 7 | 0.006 | 11 | 7 | 0.274 |
| Rheumatological disease | 5 | 5 | 0.006 | 8 | <5* | 0.456 |
| Peptic ulcer disease | <5* | | 1.000 | <5* | | 1.000 |
| Mild liver disease | 6 | <5* | 0.299 | <5* | 5 | 0.034 |
| Diabetes without chronic complications | 13 | <5* | 0.269 | 11 | 6 | 0.407 |
| Diabetes with chronic complications | | <5* | 0.136 | | <5* | 0.265 |
| Hemiplegia or paraplegia | | | | | | |
| Any malignancy, including leukemia and lymphoma | 46 | 9 | 0.531 | 32 | 20 | 0.045 |
| Moderate or severe liver disease | <5* | <5* | 0.445 | <5* | <5* | 0.059 |
| Metastatic solid tumor | <5* | <5* | 0.254 | <5* | <5* | 0.460 |
| AIDS/HIV | 7 | <5* | 0.353 | 6 | <5* | 0.705 |
| Renal disease | <5* | <5* | 0.521 | <5* | <5* | 0.612 |
| Organ transplantation | 23 | <5* | 1.000 | 13 | 13 | 0.010 |
| **Day 90 immune response** | | | | | | |
| Serological vaccine responder group, n (%) | | | | | | |
| Hypo | 27 (75.0) | 9 (25.0) | 0.006 | 16 (44.4) | 20 (55.6) | <0.001 |
| Moderate | 83 (79.0) | 22 (21.0) | | 65 (64.4) | 36 (35.6) | |
| High | 286 (89.9) | 32 (10.1) | | 247 (79.9) | 62 (20.1) | |
| Total SARS CoV-2 Spike IgG Antibodies (AU/mL ·$10^5$), median [Q1,Q3] | 2.96 [1.08,4.58] | 1.14 [0.48,2.40] | <0.001 | 3.45 [1.54,4.65] | 1.02 [0.44,2.43] | <0.001 |
| **Post day 90** | | | | | | |
| Breakthrough Infection, n (%) | | | | | | |
| Yes | 119 (87.5) | 17 (12.5) | 0.766 | 98 (73.1) | 36 (26.9) | 0.907 |
| No | 280 (85.9) | 46 (14.1) | | 232 (73.7) | 83 (26.3) | |
| Follow-up days, median [Q1,Q3] | 228 [196,258] | 264 [242,266] | <0.001 | 217 [195,257] | 250 [236,272] | <0.001 |

*Groups with small numbers (<5 participants per cell) where there is the potential that individual participants could be identified or be able to identify themselves have been edited to maintain participant confidentiality.

(Supplementary Table 1). Lastly, day 90 cellular immunity data was split into T cell responders and T cell hypo-responders. Among the 462 participants with data on Spike-specific CD4 + T cells at day 90, 399 (86.4%) were classified as responders and 63 (13.6%) as hypo-responders. Of the 449 participants with data on Spike-specific CD8 + T cells at day 90, 330 (73.5%) were classified as responders and 119 (26.5%) as hypo-responders. No significant difference in the occurrence of breakthrough infections could be observed between T cell hypo-responders and responders for neither CD4 + nor CD8 + T cells (Table 2).

Examining virus variant of the breakthrough infections, revealed that one participant was infected with an unknown variant, 10 participants with the B.1.617.2 (Delta) variant, and the remaining 156 with the B.1.1.529 (Omicron) variant. Cellular response at day 90 did not differ between participants infected with B.1.617.2 compared to B.1.1.529 (Supplementary Fig. 6b, d).

**Risk factors for cellular vaccine hypo-responsiveness at day 90.** Since age and CCI influenced the vaccine-induced cellular immune response, specific risk factors and their isolated effect on CD4 + and CD8 + cellular hypo-responsiveness were examined.

Examining participant characteristics showed that a larger proportion of cellular hypo-responders were male, older, and had a higher prevalence of comorbidities (Table 2). Further,

significantly higher levels of SARS-CoV-2 Spike IgG antibodies were observed in CD4 + T cell responders ($2.96 \cdot 10^5$ AU/mL) compared to CD4 + T cell hypo-responders ($1.14 \cdot 10^5$ AU/mL) and CD8 + T cell responders ($3.45 \cdot 10^5$ AU/mL) vs hypo-responders ($1.02 \cdot 10^5$ AU/mL) (Table 2).

Multivariable logistic regression (Fig. 5) found that although male sex, and older age (≥75 years) were associated with an

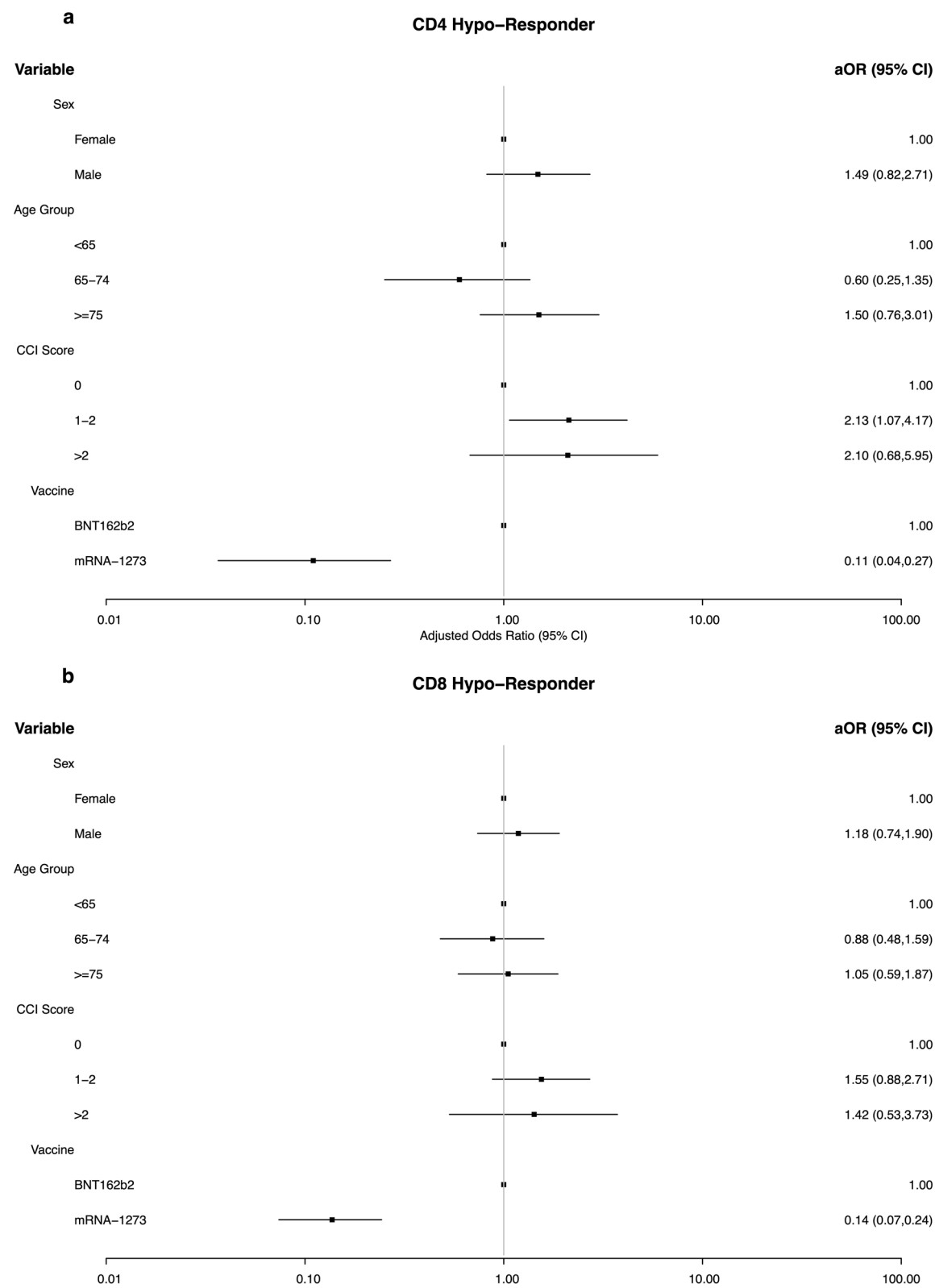

**Fig. 5 Risk factors for cellular COVID-19 vaccine hypo-responsiveness. a** Forest plot of adjusted odds ratios (aOR) for cellular CD4 + hypo-responsiveness with 95% confidence intervals (CI). Number of hypo-responders and responders, respectively, for each variable from top to bottom is; $n = 25$ and $n = 206$ (Female), $n = 36$ and $n = 186$ (Male), $n = 20$ and $n = 185$ (<65 years), $n = 11$ and $n = 123$ (65–74 years), $n = 30$ and $n = 84$ (≥75 years), $n = 36$ and $n = 306$ (CCI = 0), $n = 19$ and $n = 72$ (CCI = 1–2), $n = 6$ and $n = 14$ (CCI > 2), $n = 56$ and $n = 184$ (BNT162b2), $n = 5$ and $n = 208$ (mRNA-1273). **b** Forest plot of adjusted odds ratios (aOR) for cellular CD8 + hypo-responsiveness with 95% confidence intervals. Number of hypo-responders and responders, respectively, for each variable from top to bottom is; $n = 55$ and $n = 172$ (Female), $n = 61$ and $n = 152$ (Male), $n = 44$ and $n = 160$ (<65 years), $n = 29$ and $n = 101$ (65–74 years), $n = 43$ and $n = 63$ (≥75 years), $n = 75$ and $n = 258$ (CCI = 0), $n = 32$ and $n = 55$ (CCI = 1–2), $n = 9$ and $n = 11$ (CCI > 2), $n = 98$ and $n = 130$ (BNT162b2), $n = 18$ and $n = 194$ (mRNA-1273). Data was assessed in a multivariable logistical regression including sex, age group, Charlson Comorbidity Index (CCI) score, and vaccine type as predictors. Participants receiving ChAdOx1 as their first dose were excluded from the model given the small sample size ($n = 9$).

increased likelihood of both CD4 + and CD8 + hypo-responsiveness, these differences were not statistically significant. There also appeared to be an increased risk of hypo-responsiveness in those with more comorbidities. Participants with a CCI between 1–2 (aOR 2.13, 95%CI 1.07-4.17) and >2 (aOR 2.10, 95% CI 0.68-5.95) were over twice as likely to be CD4 + hypo-responders. A similar trend was seen for CD8 + hypo-responsiveness although the differences were again non-significant. Lastly, vaccination with mRNA-1273 was associated with lower odds of cellular hypo-responsiveness for CD4 + (aOR 0.11, 95%CI 0.04-0.27) and CD8 + (aOR 0.14, 95%CI 0.07–0.24) compared to BTN162b2.

## Discussion

In this study of 655 individuals with a high proportion of older individuals and a substantial burden of comorbidities, we report increasing proportions of SARS-CoV-2 Spike-specific CD4 + and CD8 + T cells post-vaccination in the majority of individuals. Proportions of SARS-CoV-2 Spike-specific T cells increased progressively with each vaccine dose, independently of vaccine type. However, among those with data on Spike-specific CD4 + and CD8 + T cells at day 90, 13.6% and 26.5%, respectively, were defined as hypo-responders, with males, older individuals and those with comorbidities more likely to be hypo-responders. Documented breakthrough infection occurred in roughly 30% of the cohort, of which the majority were infected with B.1.1.529 (Omicron). The breakthrough infections were equally distributed between cellular hypo-responders and responders for both CD4 + and CD8 + T cells.

This study allows the combined examination of the effect of age and CCI on T cell vaccine immunogenicity. The elderly population is at greater risk of developing severe disease, and aging has a negative impact on the ability of B cells to mount a robust immune response normally leading to production of high affinity antibodies[32–34]. Moreover, the cellular immunity of older adults is generally weaker as a consequence of immunosenescence[35,36]. Previous studies have found that elderly individuals are able to mount a cellular immune response towards SARS-CoV-2 following vaccination with an mRNA-based vaccine[37,38]. This study comprises, to our knowledge, the largest cohort of elderly individuals with comorbidities, in which the differential effect of the SARS-CoV-2 vaccine is explored. As this is the first time the novel mRNA-based vaccine platforms have been used in large trials, this is also an important evaluation parameter for future use of these platforms in other disease settings. This study found an inverse relationship between age and cellular vaccine responses evident in the CD4 + T cell compartment. The difference was of such magnitude that the proportions of Spike-specific CD4 + T cells post-second vaccine dose in the oldest age group was comparable to the pre-second vaccine dose of the youngest age group.

Considering that the cellular immune profile differs between young adults and elderly, it would be interesting to explore the

immune memory of CD4 + T cell subsets in the three age groups. Unfortunately, our AIM assay was limited to quantifying bulk responses to SARS-CoV-2 Spike and we were unable to differentiate sub-lineage memory populations. Previous studies have found both T follicular helper cells ($T_{FH}$) and memory CD4 + T cells to be efficiently recalled from convalescent and vaccinated individuals, however, these studies were conducted among younger individuals in smaller cohort studies[39–41]. Notably, the ratio of SARS-CoV-2 Spike-specific T cells reported in our study are comparable to previous studies with Spike-specific CD4 + T cells exceeding those of CD8 + T cells[17]. With increasing age, the prevalence of comorbidities also increases. When assessing the impact of comorbidities within each age group, cellular vaccine-induced immunity was progressively lower with increasing CCI. Differences were, however, not as evident in the oldest age group (≥75 years). This could be due to more pronounced consequences of immunosenescence with older age or simply a limited number of individuals in sub-groupings.

The response to SARS-CoV-2 vaccines is known to be poorer in some patient groups, as identified by multiple studies[4,42–45]. Thus, it would have been interesting to assess the impact of individual comorbidities in relation to cellular hypo-responsiveness, however this was not possible due to small subgroups (Table 1).

Since both SARS-CoV-2 spike-specific serological and cellular immunity increased following first and second doses, it was intriguing to investigate the interrelation between the two. We found that antibody high responders developed greater cellular immunity translating into 3.4-fold greater antibody levels in CD8 + T cell responders vs hypo-responders (Table 2). Lastly, our analysis allows for a study of individuals with a severely compromised antibody response. Among individuals who had very limited Spike-specific IgG vaccine responses, we detected CD4 + T cell responses in 75.0% and CD8 + T cell responses in 44.4% (Table 2). Thus, our study clearly identifies a relationship between the humoral and cellular immunity in SARS-CoV-2 vaccinated individuals. However, the study also underscores the redundancy and potential compensatory effect of the two adaptive immune components and the impact on protective immunity.

Importantly, our prospective study of SARS-CoV-2 Spike-specific T cell immunity allowed for an assessment of the impact of cellular immune memory on the occurrence of breakthrough infections. Risk of breakthrough infections with the wildtype, B.1.1.7 (Alpha) and B.1.617.2 (Delta) have been associated with decreased levels of neutralizing antibodies[15,46]. However, the effect of SARS-CoV-2 Spike-specific T cells on the risk of breakthrough infection has not, to our knowledge, been reported in humans. Our study found no differences in neither CD4 + nor CD8 + T cell responses, at day 90, in participants with breakthrough infections compared to no breakthrough infection. The majority of the breakthrough infections were caused by B.1.1.529 (Omicron; BA.1 and BA.2) but previous studies have all suggested

preserved T cell epitope responses to B.1.1.529[47–49]. Further, no difference in magnitude of T cell responses were observed between individuals infected with B.1.617.2 vs B.1.1.529. Along this line, the study by Stærke et al.[15], found no association between levels of Spike IgG antibodies and risk of B.1.1.529 breakthrough infections. Importantly, our analysis does not take into account the timing of breakthrough infections in relation to the blood sample at day 90, nor potential booster shots (third vaccine dose). Further, population transmission levels at the time of infection as well as social distancing and isolation are not considered. Collectively, all these factors weaken the predictability of protection against breakthrough infections from our data. However, considering data from animal SARS-CoV-2 challenge models, the absence of a clear protective signal in this cohort may not be surprising, as the major benefit from established CD8 + immunity appear to be on prevention of lower respiratory tract disease[50]. We were unable to infer any protective effect of the cellular immunity on disease severity as none of the breakthrough cases were hospitalized due to COVID-19 disease[15].

The present study suggests that compared to BTN162b2, vaccination with mRNA-1273 induces greater cellular immunity for both CD4 + and CD8 + T cells. An explanation for this finding could be the variation in mRNA content of the vaccines (100 µg for mRNA-1273 compared to 30 µg in BNT162b2)[23,51]. The finding that mRNA-1273 induces greater cellular immunity coincides with several previous studies showing higher antibody titers of mRNA-1273 compared to BTN162b2[52,53]. Additionally, risk factors for serological hypo-responsiveness (total SARS CoV-2 Spike IgG and ACE2 blocking antibodies) in the entire ENFORCE cohort by Søgaard et al.[4] also identified mRNA-1273 vaccine-recipients to have a lower risk of serological hypo-response. It is, however, important to note that only one case of severe COVID-19 disease with symptoms requiring hospital admission and medical treatment has been identified in the entire ENFORCE cohort. We are therefore not able to observe any differences in clinical outcome between the two mRNA vaccines[15].

Apart from vaccine type, the study found trends for increased odds for cellular hypo-responsiveness for male sex, age ≥75 years, and CCI > 0. However, apart from CCI = 1–2 in CD4 + T cell hypo-responders, none of the associations were significantly different from reference. The observation that male sex might increase the risk of cellular hypo-responsiveness is in accordance with data from the serological hypo-responsiveness study[4]. Notably, this can also be viewed in the light of the described sex bias for severe COVID-19 with higher numbers of cases, greater disease severity, and higher death rates among men than women[54,55].

Our study had some limitations, the most obvious being the skewed demographics of individuals included in the three vaccine groups caused by the temporal variations and priorities in the vaccine rollout in Denmark. Our multivariable logistic regression model adjusted for key potential confounders, however, other factors like individual comorbidities and the highly different characteristics of participants receiving BTN162b2 compared to those receiving mRNA-1273 are unaccounted for. Additionally, unknown confounding may still prevail. Lastly, some sub-groupings in our logistic regression model were quite small, which is reflected in the relatively wide confidence intervals of the aORs.

In conclusion, our study found an accordance between SARS-CoV-2 vaccine-induced cellular and serological immunity. Both compartments of the adaptive immune response increased with each vaccine dose and were progressively lower with older age and higher prevalence of comorbidities. In this large prospective cohort, we were unable to identify any threshold for protective effect of CD4 + and CD8 + Spike-specific T cell immunity against breakthrough infections. Lastly, our study characterized cellular immune hypo-responders and found risk factors for cellular hypo-responsiveness in line with findings regarding serological hypo-responsiveness. Collectively, this study contributes to a more profound understanding of cellular immunity following mRNA-based vaccination in a group with increased risk of severe COVID-19 disease.

## Data availability
Source data for the main figures can be found in Supplementary data 1. Other data is not publicly available and restricted to protect the privacy of study participants. Data from the ENFORCE cohort may be made available to researchers upon reasonable request by approval of an application to retrieve data by the ENFORCE scientific steering committee. If approval is granted data will be provided as deidentified data. The ENFORCE protocol is available at www.enforce.dk.

## Code availability
All statistical analyses and data visualization was performed using Python version 3.9 and R stats version 3.6.2 with associated packages. No source code of Python or R packages was modified. Requests for code should be directed to the corresponding author.

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

## Acknowledgements

The ENFORCE study group members all contributed substantially to the study. A full list of members of the ENFORCE study group is provided as supplementary material. We would like to thank all ENFORCE substudy participants who generously shared their time and consented to use of their data and biological samples in our study. This work was supported by the Danish Ministry of Health (document 150, parliamentary year 2020/2021, the Danish Parliament).

## Author contributions

O.S.S., J.R., N.B.S., D.R., J.L., L.Ø., and M.T. conceived and designed the study. J.L., L.Ø., O.S.S, D.R., N.B.S., and M.T. obtained funding. H.N., I.S.J., T.B., L.W., and N.B.S. provided regional oversight, recruited participants, and collected participant data. K.I., K.F., J.B., K.T.P., L.L., L.W.M., S.O.L., and I.K.H. recruited participants and provided biological material. S.D.A., A.K.H., S.R.A., E.A.M.B., R.O., L.L.D., and A.K.J. performed T cell and antibody analysis. J.R., S.F.J., and T.Ø.J. provided data extract and database merging. L.L.D., A.K.J., and J.R. analyzed the data. L.L.D., A.K.J., and M.T. wrote the manuscript. All authors provided input and approved the manuscript on behalf of the ENFORCE consortium.

## Competing interests

NBS served as principal investigator in clinical studies from Pfizer and Gilead. All other authors declare no competing interests.

## Additional information

the ENFORCE Study Group

**Sponsor** J. Lundgren[3,9,12]

**Principal Investigator** L. J. Østergaard[1,2]

**Study personnel** T. Benfield[8,9], L. Krohn-Dehli[8], D. K. Petersen[8], K. Fogh[11], E. Højmark[11], K. Iversen[11], V. Klastrup[1], F. Larsen[1], N. B. Stærke[1,2], S. Schieber[1], A. Søndergaard[1], M. Tousgaard[1], Y. Yehdego[1], J. Bodilsen[4,5], H. Nielsen[4,5], K. T. Petersen[4], M. Ruwald[4], R. K. Thisted[4], S. F. Caspersen[10], M. Iversen[10], L. S. Knudsen[10], J. L. Meyerhoff[10], L. G. Sander[10], L. Wiese[10], C. Abildgaard[6], I. K. Holden[7], I. S. Johansen[6,7], L. Larsen[6,7], S. O. Lindvig[7], L. W. Madsen[6,7] & A. Øvrehus[6]

**Scientific Steering Committee** N. A. Kruse[13], H. Lomholdt[13], T. G. Krause[14], P. Valentiner-Branth[14], B. Søborg[15], T. K. Fischer[9], C. Erikstrup[2], S. R. Ostrowski[12], H. Nielsen[4,5], I. S. Johansen[6,7], L. J. Østergaard[1,2], M. Tolstrup[1,2], N. B. Stærke[1,2], O. S. Søgaard[1,2], L. Wiese[10], T. Benfield[8,9], J. Lundgren[3,9,12] & D. Raben[3]

**Operational Group** H. Nielsen[4,5], I. S. Johansen[6,7], L. J. Østergaard[1,2], M. Tolstrup[1,2], N. B. Stærke[1,2], O. S. Søgaard[1,2], L. Wiese[10], T. Benfield[3], J. Lundgren[3,9,12], D. Raben[3], E. Jylling[16] & D. Hougaard[14]

**Coordinating Centre** S. D. Andersen[1,2], K. Lykkegaard[1], N. B. Stærke[1,2], O. S. Søgaard[1,2], M. Tolstrup[1,2] & L. J. Østergaard[1,2]

**ENFORCE Lab** S. R. Andreasen[1,2], E. Baerends[1,2], L. L. Dietz[1,2], A. K. Hvidt[1,2], A. K. Juhl[1,2], R. Olesen[2] & M. Tolstrup[1,2]

**Data and Statistical Centre** K. K. Andersen[3], W. Bannister[3], C. Bjernved[3], F. V. Esmann[3], E. Gravholdt[3], C. M. Jensen[3], S. F. Jakobsen[3], M. L. Jakobsen[3], T. Ø Jensen[3], D. Kristensen[3], J. Lundgren[3,9,12], C. Matthews[3], N. Normand[3], C. Olsson[3], D. Raben[3], J. Reekie[3] & A. Traytel[3]

[13]Lægemiddelstyrelsen, Copenhagen, Denmark. [14]Statens Serum Institut, Copenhagen, Denmark. [15]Sundhedsstyrelsen, Copenhagen, Denmark. [16]Danske Regioner, Copenhagen, Denmark.

