## [Peer Review File · Communications Medicine]

Reviewers' comments:

Reviewer #1 (Remarks to the Author):

In this manuscript authors have assessed longitudinally the effect of age, sex, and comorbidities on the T cell response to two mRNA vaccines in cohorts of vaccinees mostly >65 years of age. Additionally, authors correlate the magnitude of T cell responses to breakthrough infections in the elderly population. They find that: (1) T cell responses increase in magnitude following primary and booster vaccinations; (2) Reduced CD4 T cell and antibody responses were detected in cohorts >75 years of age at 90 days after vaccination; (3) No statistically significant differences in CD4 T cell responses were found between samples from vaccinees with or without co-morbidities; (4) Antibody levels were lower in vaccinees >75 years of age with >2 co-morbidities; (5) Higher CD4 T cell responses were associated with greater antibody levels; (6) Breakthrough infections were not linked to differences in T cell responses.

Overall, this is a very significant study to assess T cell responses to vaccines in elderly statesman with and with comorbidities. Because none of the vaccinees developed severe disease after a breakthrough infection, authors were unable to correlate levels of T cell immunity and protection from severe disease in older vaccinees with and without comorbidities. It is not surprising that breakthrough infections are not linked to T cell immunity because T cells do not confer sterilizing immunity, unlike antibodies.

The impact of this study would be greater if the following data is included in the manuscript:

1. How does antibody levels relate to breakthrough infections? Authors need to provide neutralizing antibody titers!
2. Data on T cell function (cytokine production) would have helped in understanding the effect of age and/or comorbidities on vaccine-induced cellular immunity to SARS-CoV-2.
3. Inclusion of T cell and antibody responses of a younger cohort would be essential to compare the effect of age on the magnitude of antibody and T cell responses.

Reviewer #3 (Remarks to the Author):

In the present manuscript, the degree of anti-Spike (S) IgG response (= humoral immunity) and CD4/CD8 T-cell response (= cellular immunity) was investigated after one (day 21) and two (day 90) vaccinations, respectively. Only a few first vaccinations were given with a vector vaccine, all second vaccinations were given with mRNA-based vaccines. IgG levels and the proportion of S-specific T cells were highest after the second vaccination. The type of mRNA vaccine, male sex, age 75 years and older, and the presence of comorbidities were associated with lower cellular immunity. It could not be clarified what level of T-cell immunity is required for protection against breakthrough infection.

The study is solidly done and the illustrations are clear and meaningful.

However, some major comments should be made:

- 1) The authors should state from which SARS-CoV-2 lineage the S-protein used for T-cell

stimulation was derived. It should also be briefly stated which tubes were used for the collection of whole blood samples and how the PBMCs were prepared.

2) The authors should also indicate which assay was used to measure the anti-S IgGs. It is not sufficient to refer to a publication here.

3) Looking at Figure 5, there is actually only a clear effect for the type of mRNA vaccine. For the other factors, which according to the authors should influence the T-cell response, the effect, if present at all, is marginal (inclusion of 1 in the CI) and is not shown in the figure for a CCI>0. According to the authors, the type of mRNA vaccine makes no difference in clinical outcome. However, I cannot find any data on clinical equivalence.

4) The problem of breakthrough infections with omicron is of course also due to the pronounced immune escape of this variant. Ultimately, it is a serotype of its own, clearly distinguishable from the pre-VOCs and alpha, beta, gamma and delta variants. An immune response against omicron can only be measured after a third vaccination, but this was not the subject of the present study. There are several publications on the antibody and T-cell response to omicron. In this respect, I think it is questionable whether valid statements can be made about the threshold value of vaccine-induced (antigen-derived from pre-VOC) protective cellular immunity on the basis of the breakthrough infections that occurred predominantly with omicron. Furthermore, the authors would have to define what they mean by protective T-cell immunity.

Reviewer #4 (Remarks to the Author):

In this study the authors assess humoral and cellular antigen specific responses following SARS-CoV-2 vaccination and their association to breakthrough infections. Sample size is very large, manuscript is well written and while the study is reasonably well conducted there is several issues which need to be addressed prior to publication.

The authors did not convince me that they have any true non responders in their cohort. The authors did not detect response by the AIM assay but that is not the only method to measure antigen specific T cell response. There is evidence that IFN- γ ELISPOT is one of the most sensitive methods for measuring antigen specific T cell response so the authors should test at least a subset of "non responders" to see if these are truly non responders or their method is just not sensitive enough. If they don't have technical capabilities for ELISpot they can measure IFN-gamma by flow cytometry.

It is also unclear how the gated of positivity were set in flow cytometric analysis of expression of CD69, OX-40 and 4-1BB. These are all markers which require fluorescence minus one controls for proper gating. Did the author use such controls?

Do the authors have the information on absolute lymphocyte counts and cell viability in their groups? Without it (especially absolute cell counts) the data is hard to interpret. For example, percentage of activated CD4 T cells was significantly lower in the oldest group even at day 0 prior to vaccination.

The authors should not overstate their results and should emphasize in the abstract that they observed a decrease with age only in CD4 T cell response but not CD8.

The authors should properly cite previous similar studies which were assessing vaccination induced antigen specific SARS-CoV-2 responses in the context of age such as Collier et al. Nature 2021 and Jergovic et al. Nature Communications 2022.

Point-by point response to Reviewers' comments:

Reviewer #1

In this manuscript authors have assessed longitudinally the effect of age, sex, and comorbidities on the T cell response to two mRNA vaccines in cohorts of vaccinees mostly >65 years of age. Additionally, authors correlate the magnitude of T cell responses to breakthrough infections in the elderly population. They find that: (1) T cell responses increase in magnitude following primary and booster vaccinations; (2) Reduced CD4 T cell and antibody responses were detected in cohorts >75 years of age at 90 days after vaccination; (3) No statistically significant differences in CD4 T cell responses were found between samples from vaccinees with or without comorbidities; (4) Antibody levels were lower in vaccinees >75 years of age with >2 comorbidities; (5) Higher CD4 T cell responses were associated with greater antibody levels; (6) Breakthrough infections were not linked to differences in T cell responses.

Overall, this is a very significant study to assess T cell responses to vaccines in elderly statesman with and with comorbidities. Because none of the vaccinees developed severe disease after a breakthrough infection, authors were unable to correlate levels of T cell immunity and protection from severe disease in older vaccinees with and without comorbidities. It is not surprising that breakthrough infections are not linked to T cell immunity because T cells do not confer sterilizing immunity, unlike antibodies.

The impact of this study would be greater if the following data is included in the manuscript:

1. How does antibody levels relate to breakthrough infections? Authors need to provide neutralizing antibody titers!

Stratifying participants by breakthrough infection status (no n=150; yes n=351), show no significant difference in SARS-CoV-2 Spike IgG (AU/mL) at day 90 (p=0.278):

The ENFORCE study group, which we are a part of, has also published data on SARS-CoV-2 Spike IgG (AU/mL) in relation to breakthrough infection with Omicron within the entire ENFORCE cohort (close to 7000 participants) and found that the quantitative level of anti-spike IgG have limited impact on the risk of breakthrough infection with Omicron (<https://doi.org/10.1038/s41467-022-32254-8>).

2. Data on T cell function(cytokine production) would have helped in understanding the effect of age and/or comorbidities on vaccine-induced cellular immunity to SARS-CoV-2.

This is a great point that would have been interesting to explore following SARS-CoV-2 Spike stimulation. Importantly, upregulation of the AIM markers following peptide recognition through the T cell receptor provides a broader and more sensitive perspective on T-cell activation than for instance intracellular cytokine stain (ICS).

Multiple studies have shown that the AIM assay is superior to classical methods quantifying cytokine production in identifying antigen-specific T cells. The classical methods often fail to identify the total pool of antigen-specific T cells as they are biased to the cytokine panels used, which are often Th1-skewed (IFN- γ and IL-2).
<https://doi.org/10.1371/journal.pone.0186998>
[doi:10.3390/vaccines6030050](https://doi.org/10.3390/vaccines6030050)

We do unfortunately not have supernatants stored from these peptide simulations thus evaluating cytokine secretion following stimulation is not manageable in the scope of this publication.

3. Inclusion of T cell and antibody responses of a younger cohort would be essential to compare the effect of age on the magnitude of antibody and T cell responses.

We agree studying a younger cohort to compare these findings would be very interesting. The ENFORCE study was designed to primarily enroll elderly and vulnerable participants with comorbidities in high risk of severe COVID-19 disease. Furthermore, as the vaccine coverage in Denmark is about 87% of all above 18 years of age it would be quite difficult to collect material from a younger cohort interested in participating in a vaccine research project at this time point.

Reviewer #3

In the present manuscript, the degree of anti-Spike (S) IgG response (= humoral immunity) and CD4/CD8 T-cell response (= cellular immunity) was investigated after one (day 21) and two (day 90) vaccinations, respectively. Only a few first vaccinations were given with a vector vaccine, all second vaccinations were given with mRNA-based vaccines. IgG levels and the proportion of S-specific T cells were highest after the second vaccination. The type of mRNA vaccine, male sex, age 75 years and older, and the presence of comorbidities were associated with lower cellular immunity. It could not be clarified what level of T-cell immunity is required for protection against breakthrough infection.

The study is solidly done and the illustrations are clear and meaningful.

However, some major comments should be made:

1. The authors should state from which SARS-CoV-2 lineage the S-protein used for T-cell stimulation was derived. It should also be briefly stated which tubes were used for the collection of whole blood samples and how the PBMCs were prepared.

We have added a section in the methods describing PBMC isolation from CPT tubes (**p. 18 lines 418-428**).

“Whole blood was collected in sodium citrate/Ficoll blood collection tubes from BD Vacutainer (BD CPT, Cat. No.: BDAM362782). PBMCs were isolated from three CPT tubes per participant. CPT tubes were centrifuged at 1500xg for 20 mins within 2 hours of blood collection. Centrifuged CPT tubes were reverted slowly, the supernatant of all three CPT tubes were pooled and centrifuged at 400xg for 10 mins. The supernatant was then discarded and PBMCs were resuspended and washed in PBS containing 2% FBS. PBMCs were pelleted by centrifugation at 400xg for 10 mins and resuspended in media for cryopreservation (FBS with 10% DMSO). Immediately following resuspension in media with DMSO, cells were placed in freezing containers and cryopreserved at -80°C for at least 24 hours, subsequently the cells were moved to -150°C for long-term storage.”

Further, we have stated the lineage of the S-protein used for T cell stimulations in the methods section “SARS-CoV-2 Spike-specific T Cells” (**p. 19 lines 435-439**).
“Purified PBMCs were stimulated with PepMix™ SARS-CoV-2 (JPT peptides product code PM-WCPV-S-2) at 2µg/ml or negative control (Dimethyl sulfoxide) for 20 hours. The PepMix™ contains a pool of 315 (158+157) peptides derived from a peptide

scan (15mers with 11 aa overlap) through the Spike glycoprotein (Swiss-Prot ID: P0DTC2) of SARS-CoV-2 (Wuhan-Hu-1 lineage)."

2. The authors should also indicate which assay was used to measure the anti-S IgGs. It is not sufficient to refer to a publication here.

We have added the following description to the methods section "SARS-CoV-2 Antibody Profiling" (**p. 20 lines 461-467**).

"Serum levels of SARS-CoV-2 Spike IgG antibodies were measured at all study visits using the MesoScale Diagnostic Multiantigen Serology Assay. Plates were added samples diluted 1:5000 and a four-fold seven-point dilution of the MSD standard reference and a blank in diluent buffer. Plates were read on a MESO SECTOR S600 Reader. Data analysis was performed utilizing the MSD Discovery Workbench Software (Version 4.0). Total IgG concentrations were calculated by fitting the electro chemiluminescence signals to the calibration curves".

3. Looking at Figure 5, there is actually only a clear effect for the type of mRNA vaccine. For the other factors, which according to the authors should influence the T-cell response, the effect, if present at all, is marginal (inclusion of 1 in the CI) and is not shown in the figure for a CCI>0. According to the authors, the type of mRNA vaccine makes no difference in clinical outcome. However, I cannot find any data on clinical equivalence.

We agree that the results in the abstract should be rephrased. We have re-written the following section in abstract:

"Male sex, age group ≥ 75 years, and CCI > 0 was associated with an increased likelihood of being a cellular hypo-responder while vaccine type was a significant risk factor." (p. 2 lines 46-48).

We apologize for not being clear about the term clinical outcome in the context of this manuscript. The clinical endpoint/clinical outcome refers to hospitalizations caused by a SARS-CoV-2 infection as we do not have detailed info on disease outcome if study participants do not require medical assistance. As described in Stærke et al. (<https://doi.org/10.1038/s41467-022-32254-8>), only one case of severe COVID-19 disease with symptoms requiring hospital admission and medical treatment has been identified in the entire ENFORCE cohort. We are therefore not able to observe any clinical differences (i.e. differences in the clinical endpoint of hospitalization) between the two mRNA vaccines. This has been clarified in the manuscript:

"It is, however, important to note that only one case of severe COVID-19 disease with symptoms requiring hospital admission and medical treatment has been identified in the entire ENFORCE cohort. We are therefore not able to observe any differences in clinical outcome between the two mRNA vaccines" (p. 14 lines 319-323).

4. The problem of breakthrough infections with omicron is of course also due to the pronounced immune escape of this variant. Ultimately, it is a serotype of its own, clearly distinguishable from the pre-VOCs and alpha, beta, gamma and delta variants. An immune response against omicron can only be measured after a third vaccination, but this was not the subject of the present

study. There are several publications on the antibody and T-cell response to omicron. In this respect, I think it is questionable whether valid statements can be made about the threshold value of vaccine-induced (antigen-derived from pre-VOC) protective cellular immunity on the basis of the breakthrough infections that occurred predominantly with omicron. Furthermore, the authors would have to define what they mean by protective T-cell immunity.

We thank the reviewer for this very valid point that the vaccine-induced cellular immunity examined in the present study is derived from the original Wuhan-Hu-1 lineage and the majority of breakthrough infections in this study is caused by the Omicron variant. We also agree that Omicron is a VOC clearly distinguishable from the previous VOCs. However, with regards to T cell epitopes, Omicron retains a high degree of unmutated T cell epitopes compared to previous VOCs.

Choi SJ et al. (doi: 10.1038/s41423-022-00838-5) show high preservation of CD8+ T cell epitopes in the Spike protein of Omicron compared to Wuhan-Hu-1 (with 88.4 % preservation of T cell epitopes in the Spike protein of the Omicron variant).

Similarly, Tarke A et al. (doi: 10.1016/j.cell.2022.01.015) show that 84% (CD4+) and 85% (CD8+) of T cell responses were preserved against Omicron on average by AIM assay and they report that there are no differences in T cell responses (measured by the AIM assay) against VOC-specific peptide stimulation.

Based on this T cell epitope conservation in the Omicron variant and the comparative data, we believe vaccine-induced protective cellular immunity (antigen-derived from vaccine strain) can indeed be used to study breakthrough infections with Omicron.

With regards to protective T cell immunity, the “protective T cell immunity threshold” was a poor choice of words in our abstract. Our study aimed to examine differences in day 90 vaccine-induced T cell immunity between cases (breakthrough infection) and controls (no breakthrough infection) to identify potential protective effects of T cell immunity on breakthrough infections. However, the study was unable to identify any protective effect of T cell immunity. The phrasing is edited in the abstract: *“Assessing breakthrough infections, no protective effect of T cell immunity could be identified” (p. 2 line 46).*

Reviewer #4

In this study the authors assess humoral and cellular antigen specific responses following SARS-CoV-2 vaccination and their association to breakthrough infections. Sample size is very large, manuscript is well written and while the study is reasonably well conducted there is several issues which need to be addressed prior to publication.

1. The authors did not convince me that they have any true non responders in their cohort. The authors did not detect response by the AIM assay but that is not the only method to measure antigen specific T cell response. There is evidence that IFN- γ ELISPOT is one of the most sensitive methods for measuring antigen specific T cell response so the authors should test at least

a subset of “non responders” to see if these are truly non responders or their method is just not sensitive enough. If they don't have technical capabilities for ELISpot they can measure IFN-gamma by flow cytometry.

We apologize for not being clearer on our intention of using the descriptor “cellular non-responder”. Our definition of a cellular non-responder status is based on a threshold value set as the median+1xSD of SARS-CoV-2 Spike-specific T cells at baseline (day 0) in the entire ENFORCE substudy cohort.

We used this threshold as a conservative way of evaluating a true AIM signal from the background signal in our assay. We agree that our term non-responder is ripe for misunderstanding as we did not intend to describe a de facto cellular non-response but rather a low or absent signal that could be detected using the AIM assay. Therefore, by categorizing a participant as a “non-responder” we do not claim that this person does not respond to vaccination. Therefore, to avoid any misconceptions, we have changed “non-responder” to “hypo-responder” throughout the manuscript and used the stratification of study participants into the two groups (hypo-responders and responders) to assess the cellular responsiveness with the AIM assay, and define characteristics of participants with a low -or absent AIM response (hypo-responders) vs. a higher AIM response (responders).

We have now included the following sentence describing this in the results section “Serological and Cellular Vaccine Responder Group”:

“The study participants were stratified into cellular vaccine responder groups (hypo-responders and responders) to assess the cellular responsiveness with the AIM assay, and define characteristics of participants with a low- or absent AIM response (hypo-responders) vs. a higher AIM response (responders)” (p. 20 lines 474-477).

2. It is also unclear how the gates of positivity were set in flow cytometric analysis of expression of CD69, OX-40 and 4-1BB. These are all markers which require fluorescence minus one controls for proper gating. Did the author use such controls?

FMO controls were used in setting up the compensation matrix. We found that the background fluorescence of FMO controls were lower than in our full stain setup and they were therefore not used to set the gates of positivity. Instead, negative controls were used to set the gates of positivity.

3. Do the authors have the information on absolute lymphocyte counts and cell viability in their groups? Without it (especially absolute cell counts) the data is hard to interpret. For example, the percentage of activated CD4 T cells was significantly lower in the oldest group even at day 0 prior to vaccination.

Thank you for your insight, we agree that information of cell counts and cell viability is crucial for data interpretation.

With our flow cytometry assay it is not possible to provide an absolute lymphocyte count. The AIM assay reports a relative proportion of SARS-CoV-2 Spike-specific T cells and not a proportion of the absolute number of lymphocytes specific for SARS-

CoV-2 Spike. The samples collected for this study was not subjected to differential leucocyte count and hence we do not have access to data on absolute T cell counts.

As described in the methods section, two measures are performed at data analysis to ensure proper quality of the data. First, samples with a viability below 70 % at flow acquisition (using the viability stain) are not included in data analysis to minimize false positive signal from autofluorescent dead cells. Second, samples in which we are unable to acquire greater than 10,000 events are also excluded from the data analysis. This is done to ensure at least around 100 events in our end gates (CD69+, OX-40+ and 4-1BB+) as the proportion of cells positive for our AIMs is often around 1% (1% of 10,000 = 100).

These two measures have stringently been used across the data analysis.

4. The authors should not overstate their results and should emphasize in the abstract that they observed a decrease with age only in CD4 T cell response but not CD8.

This is a very valid point that we have addressed throughout the manuscript and particularly in the abstract where we have edited the sentence to the following: *“Reduced serological immunity and frequency of CD4+ Spike-specific T cells was observed in the oldest age group and higher Charlson Comorbidity Index (CCI) categories.”* (p. 2 lines 43-45).

5. The authors should properly cite previous similar studies which were assessing vaccination induced antigen specific SARS-CoV-2 responses in the context of age such as Collier et al. Nature 2021 and Jergovic et al. Nature Communications 2022.

We were not aware of these publications. They are indeed very fitting and have now been cited in our final manuscript by adding the following to the discussion section. Thank you for the suggestions.

“Previous studies have found that elderly individuals are able to mount a cellular immune response towards SARS-CoV-2 following vaccination with an mRNA-based vaccine (Jergovic, Collier)”. (p. 11 lines 237-240)

Reviewers' comments:

Reviewer #1 (Remarks to the Author):

In the previous review, one of my concerns was: 1. How do antibody levels relate to breakthrough infections? Authors need to provide neutralizing antibody titers!

Authors have provided IgG levels in the rebuttal letter. It is unclear how the IgG levels were quantified? Using the Wuhan Spike or the Omicron spike as antigen? This is important because breakthrough infections are caused by OMICRON.

Reviewer #3 (Remarks to the Author):

The authors have discussed all the reviewers' comments and, where necessary, also incorporated them into the revised manuscript. In this respect, I think that the revised manuscript has been improved once again compared to the already scientifically sound original version.

Reviewer #4 (Remarks to the Author):

The authors have successfully addressed all my comments.

Rebuttal letter - Reviewers' comments:

Reviewer #1 (Remarks to the Author):

In the previous review, one of my concerns was: 1. How do antibody levels relate to breakthrough infections? Authors need to provide neutralizing antibody titers!

Authors have provided IgG levels in the rebuttal letter. It is unclear how the IgG levels were quantified? Using the Wuhan Spike or the Omicron spike as antigen? This is important because breakthrough infections are caused by OMICRON.

We agree it is important for analysis whether IgG levels are quantified using Wuhan Spike or Omicron Spike and hope the following will clarify. The IgG levels were quantified using the Wuhan-Hu-1 spike antigen, thus "based on Wuhan-Hu-1 strain Spike antigen" has been added to the methods section as follows:

SARS-CoV-2 Antibody Profiling

Serum levels of SARS-CoV-2 Spike IgG antibodies were measured at all study visits using the MesoScale Diagnostic Multiantigen Serology Assay **based on Wuhan-Hu-1 strain Spike antigen**. Plates were added samples diluted 1:5000 and a four-fold seven-point dilution of the MSD standard reference and a blank in diluent buffer. Plates were read on a MESO SECTOR S600 Reader. Data analysis was performed utilizing the MSD Discovery Workbench Software (Version 4.0). Total IgG concentrations were calculated by fitting the electro chemiluminescence signals to the calibration curves.

Reviewer #3 (Remarks to the Author):

The authors have discussed all the reviewers' comments and, where necessary, also incorporated them into the revised manuscript. In this respect, I think that the revised manuscript has been improved once again compared to the already scientifically sound original version.

Reviewer #4 (Remarks to the Author):

The authors have successfully addressed all my comments.

REVIEWERS' COMMENTS:

Reviewer #1 (Remarks to the Author):

Authors still ignored the request to provide antibody neutralisation titers!

Reviewer #3 (Comments on the Report of Reviewer #1)

The Omicron line forms its own serotype, which is clearly different from the Wuhan line and the previous VOCs. In this respect, it is to be expected that vaccinees who were immunised with the previous mRNA vaccines will build up neutralising antibodies against Omicron BA.1 and BA.2 (the lines that circulated until the end of the study), but the titer will be very low. Not without reason, mRNA vaccines adapted to Omicron have been developed and have been in use since the end of the summer. The IgG test applied uses an antigen derived from Wuhan virus. Commercial IgG tests using antigens from Omicron BA.1 or BA.2 or even BA.4 and BA.5 are not available, I believe. It would be expected that the antibody concentrations (not titres!) would then be significantly lower.

The neutralisation tests required by the reviewer assume that the authors have a safety level 3 laboratory and different SARS-CoV-2 isolates, ideally wild type, Delta and Omicron BA.1 and BA.2. This is a very big effort and at the end there will be an expected result (high titres against wild type, lower titres against Delta, low titres against BA.1/BA.2). Perhaps the authors can discuss this aspect a little in the light of the currently available literature data. Since the focus of the paper is primarily on cellular immunity anyway, this should be sufficient in my view.

Rebuttal letter - Reviewers' comments:

Reviewer #1 (Remarks to the Author):

Authors still ignored the request to provide antibody neutralization titers!

It was never our intention to ignore any comments. Unfortunately, we are not able to provide neutralizing titers on the 655 participants of this study. In this large prospective cohort study, the risk of breakthrough infection in relation to SARS-CoV-2 anti-spike IgG levels has been assessed in the entire cohort, constituting 6076 participants, (these data have been published in the paper by Nina Breinholt Stærke et al (<https://doi.org/10.1038/s41467-022-32254-8>)). In this paper, we reported that the quantitative level of anti-spike IgG have limited impact on the risk of breakthrough infection with Omicron. As requested from the editor, we will address the concern of neutralization in our discussion with respect to currently available literature as suggested by reviewer #3.

Reviewer #3 (Comments on the Report of Reviewer #1):

The Omicron line forms its own serotype, which is clearly different from the Wuhan line and the previous VOCs. In this respect, it is to be expected that vaccinees who were immunised with the previous mRNA vaccines will build up neutralising antibodies against Omicron BA.1 and BA.2 (the lines that circulated until the end of the study), but the titer will be very low. Not without reason, mRNA vaccines adapted to Omicron have been developed and have been in use since the end of the summer. The IgG test applied uses an antigen derived from Wuhan virus. Commercial IgG tests using antigens from Omicron BA.1 or BA.2 or even BA.4 and BA.5 are not available, I believe. It would be expected that the antibody concentrations (not titres!) would then be significantly lower.

The neutralisation tests required by the reviewer assume that the authors have a safety level 3 laboratory and different SARS-CoV-2 isolates, ideally wild type, Delta and Omicron BA.1 and BA.2. This is a very big effort and at the end there will be an expected result (high titres against wild type, lower titres against Delta, low titres against BA.1/BA.2). Perhaps the authors can discuss this aspect a little in the light of the currently available literature data. Since the focus of the paper is primarily on cellular immunity anyway, this should be sufficient in my view.

Indeed, the Omicron line forms its own serotype clearly different from the Wuhan line and we agree that the neutralizing titers would be expected to be lower for BA.1 and BA.2 relative to the original Wuhan strain. To accommodate the request of a brief discussion on this manner, we have included the following in our discussion:

“We did not assess antibody neutralization ability to block B.1.1.259 infection in these cases. Several studies have found reduced neutralizing titers of Spike IgG antibodies against evolving variants in the B.1.1.529 lineage. However, following a third vaccine dose, the neutralizing ability increased significantly across all variants of concern (41-44)”